# Proposing Necessary but Not Sufficient Conditions Analysis as a Complement of Traditional Effect Size Measures with an Illustrative Example

**DOI:** 10.3390/ijerph19159402

**Published:** 2022-07-31

**Authors:** Ana M. Greco, Georgina Guilera, Laura Maldonado-Murciano, Juana Gómez-Benito, Maite Barrios

**Affiliations:** 1Departament de Psicologia Social i Psicologia Quantitativa, Facultat de Psicologia, Universitat de Barcelona, 08035 Barcelona, Spain; gguilera@ub.edu (G.G.); lmaldonado@ub.edu (L.M.-M.); juanagomez@ub.edu (J.G.-B.); mbarrios@ub.edu (M.B.); 2Estudis de Dret i Ciència Política, Universitat Oberta de Catalunya (UOC), Rambla del Poblenou, 156, 08018 Barcelona, Spain; 3Grup d’Estudis d’Invariància de la Mesura i Anàlisi del Canvi en els Àmbits Social i de la Salut (GEIMAC), Institut de Neurociències (UBNeuro), Universitat de Barcelona, 08035 Barcelona, Spain

**Keywords:** necessary condition analysis, NCA, effect size, measure, interpretation

## Abstract

Even though classic effect size measures (e.g., Pearson’s r, Cohen’s d) are widely applied in social sciences, the threshold used to interpret them is somewhat arbitrary. This study proposes necessary condition analysis (NCA) to complement traditional methods. We explain NCA in light of the current limitations of classical techniques, highlighting the advantages in terms of interpretation and translation into practical terms and recognizing its weaknesses. To do so, we provide an example by testing the link between three independent variables with a relevant outcome in a sample of 235 subjects. The traditional Pearson’s coefficient was obtained, and NCA was used to test if any of the predictors were necessary but not sufficient conditions. Our study also obtains outcome and condition inefficiency as well as NCA bottlenecks. Comparison and interpretation of the traditional and NCA results were made considering recommendations. We suggest that NCA can complement correlation analyses by adding valuable and applicable information, such as if a variable is needed to achieve a certain outcome level and to what degree.

## 1. Introduction

Recent arguments about how we use, report, and interpret measurements to test the link between two variables are reframing statistical practices in social sciences [1,2,3]. The strength of association between a dependent and an independent variable is usually measured using classical effect sizes such as Pearson’s r. These coefficients are often interpreted using the classic Cohen’s criteria, which consider a correlation coefficient r as small when r is between 0.10 and 0.29, medium when r is between 0.30 and 0.49, and large when r is equal to or larger than 0.50 [4]. However, in a posterior edition, the same author warned that this threshold should only be used in the lack of a better reference framework [5]. The principle of applying a standard threshold to very different variables, relationships, and contexts has been recently qualified as “nonsense” [1] and “arbitrary” [3].

A recent study based on the 708 correlations of individual differences and several outcomes (e.g., narcissism, self-esteem, political orientation) obtained from 87 meta-analyses found that less than 3% of the studies reported an *r* coefficient larger than 0.50 and that the 25th, 50th, and 75th percentiles corresponded to *r* coefficients of 0.11, 0.19, and 0.29, respectively [2]. Another study summing up the results of 322 meta-analyses of psychosocial phenomena (e.g., women experiencing more anxiety than men) revealed that the majority of the effects found were around *r*~0.21, and only 5.3% of all effects were larger than 0.50 [6]. Finally, a meta-analysis comparing the predictive validity of personality traits with that of cognitive ability and socioeconomic status on three important outcomes (i.e., mortality, divorce, and occupational attainment) found correlation coefficients ranging from 0.02 to 0.24 [7].

All this evidence suggests that Cohen’s criteria seem inapplicable (or arguably applicable) in the framework of the social sciences. If the main conclusions found through research in social sciences rely on these coefficients and thresholds, we risk arriving at misleading inferences, which might lead to unwanted consequences. For instance, we could conclude that extraversion has a small effect on risky sexual behaviour because a recent meta-analysis [8] reported an overall *r* = 0.18 between these two variables, when in fact, the value is very close to the average effect size found in personality traits concerning other similar outcomes [2,9]. Some authors suggest that the criteria for interpreting a correlation coefficient should be based on benchmarks from classic or meta-analytic studies of the referenced framework rather than on a standard or arbitrary cut-off point [1,2]. This would make the coefficient more meaningful to the context in which it should be understood. Another option would be to compare the correlation coefficient to an intuitively understood non-psychological relationship [1], such as the relieving effect of drugs such as ibuprofen to combat headaches (*r* = 0.11) or the tendency of men to weigh more than women (*r* = 0.26). Other proposals are converting effect sizes into understandable consequences of the phenomenon. Examples of these are the two one-sided tests (TOST) to reject the presence of the smallest effect size of interest (SESOI) if the effect is equivalent to or lower than the actual effect [10] or the binominal effect-size display (BESD) [11].

In addition, qualifying effects as “negligible” or “indistinguishable” when the associated *r* coefficient is below 0.20 may be overlooking the cumulative or multiplicative effect [1]. For instance, the link between a baseball player’s outcome in a single at-bat and his overall batting mean being *r* = 0.05 can be considered small, but if we consider that the total bats per season are 550, the cumulative effect becomes more relevant [12]. The effect of a growth mindset on students’ achievement was considered as weak in a meta-analysis that found an *r* = 0.08 (Sisk et al., 2018), but translating this to a relative increase in grade point average (GPA) can lead to a considerable effect, since it multiplies by the total number of students at a whole school district or college campus.

In summary, researchers in social sciences urgently need new ways to interpret and report their analysed data in meaningful ways. The current suggestions are to change the threshold [2], establish a meaningful reference framework [3], translate coefficients into understandable comparisons [1], complement the indicators with additional tests [10], or even to use all of the aforementioned strategies, when applicable. We propose necessary condition analysis (NCA) [13] as a complementary method to the classic indicators used. This method provides applicable and meaningful estimates that can be easily translated into practical effects. Here, we aim to describe the method and provide an illustrative example to analyze how it can give an alternative to the limitations mentioned above.

### 1.1. Necessary Condition Analysis (NCA)

NCA aims to check if an independent variable is a necessary but not sufficient condition for an outcome. This means that when the values of the independent variable are low or zero, the outcome cannot occur. However, if the independent variable is present to a certain degree, the outcome becomes possible, subject to other conditions. In this context, it is important to distinguish the concept of “necessity” from that of “sufficiency” [14,15]. Whereas traditional methods such as Pearson’s *r*, Cohen’s *d,* or regression coefficients test if a predictor is sufficient (on average) to increase or decrease the presence of a particular outcome, NCA assesses the necessity of a predictor by testing whether an outcome can occur in the absence of this variable. Examples of research questions and hypotheses that can be answered through “sufficiency” or “necessity” logic are shown in Figure 1.

The advantage of NCA over traditional methods is to identify the degree to which a variable must be present for the outcome to reach a certain level [18]. For instance, in a recent study [15], NCA was used among other predictors to assess whether students’ attendance percentage was a necessary but not sufficient condition for academic achievement, as measured by their GPA. Authors hypothesized that attending a certain proportion of classes allowed students to achieve the desired grade, although this would not guarantee such an outcome, as it also depends on other variables. Using the NCA, they concluded that attending at least half of the classes was required to reach a grade of 90%. Some students with grades lower than 70% reported almost perfect attendance, suggesting that attendance is necessary but not sufficient to achieve a particular grade level. 

NCA has also been applied to assess whether human resources interventions are needed to achieve a certain degree of performance [14], how much intelligence is necessary to display high levels of creativity [17,19], what degree of workplace spirituality is needed to reach high levels of job satisfaction, commitment, and work–life balance [20], and which personality traits make it possible to avoid certain impulsive behaviours [16]. These studies represent examples of how this type of analysis can complement traditional techniques, providing meaningful insights into relevant research questions in different social sciences fields. 

To estimate to what extent an independent variable X is necessary for certain levels of an outcome Y, NCA uses scatterplots. If variable X is a necessary condition for Y, the scatterplot shows an area without any observations in the upper-left corner, the ceiling zone. This empty area represents the amount of X that is indispensable (but not enough) for Y to occur. Hence, the ceiling zone is interpreted as how much the independent variable constrains possible outcome levels. This zone is limited by the ceiling line, which can be drawn in different ways according to the assumptions of the individual variables and how they are distributed [13]. The empirical scope of the minimum and maximum observed values of X and Y define the total area for which observations are possible. An effect-size measure of the necessary condition can be computed by dividing the ceiling zone area by the empirical scope area. This is represented by the letter *d,* which can be considered small when *d* < 0.1, medium when values are between 0.1 and 0.3, and large when *d* > 0.3 [13]. Other coefficients called bottlenecks can also be obtained to illustrate how much of each independent variable is necessary to reach a particular outcome level. They can be calculated using the cut-off point of interest according to the outcome tested and its scale. For instance, we can establish the percentage of class attendance needed to achieve a GPA above the 75th percentile [15]. 

Of course, not all values of the independent variable restrict the outcome. Inversely, not all outcome levels can be limited by the independent variables. In the NCA framework, these limited levels are called inefficiencies [21]. Outcome and condition inefficiencies could be understood respectively as the cut-off points from which Y cannot be constrained by X and the level of X that is not needed even for the highest possible value of Y. Following the previous example, outcome inefficiency would represent the minimum level of GPA that can be reached even without any specific percentage of class attendance, and a percentage above the condition of inefficiency for class attendance would not be needed to reach the highest possible level of GPA. 

A significance test can be performed to test whether d differs from an effect size calculated with random data [22]. This permutation test combines observations of X with observations of Y randomly until the number of observations in the sample is reached. Then, the operation is repeated 10,000 times. The p-value can be interpreted using traditional thresholds and is the sum of samples that obtained an effect size greater or equal to the one found by the NCA, divided by the total samples generated.

As the use of NCA grows among social scientists [14], so do the critiques. For instance, a recent study found an increased type I error rate in large samples and a tendency for the significance of the necessary effect to escalate with higher degrees of sufficiency [23]. This link between sufficiency and necessity has also been questioned by an article warning about causal inferences drawn from these analyses [24]. It is also true that some scatterplots resulting from the distribution of two completely unrelated variables may show an empty space, and NCA would not be able to distinguish these cases from an actual necessary but not sufficient relation [25]. Finally, another critical point may be the use of the standard cut-off points proposed by the original author [13] without considering the evidence in which the research question is framed, a limitation that also applies to traditional effect size measures [1]. 

In the following example, we provide an illustrative application of NCA and comment on how it can be a complement to traditional effect size coefficients. We also consider its possible drawbacks, provide considerations for its use to minimize the effect of these limitations, and propose future research lines. 

### 1.2. Illustrative Example: Is Academic Performance Possible without Previous Achievement, Time Spent Studying, and Conscientious Personality Traits?

To illustrate the use of NCA, we test if any of the predictors found to significantly contribute to academic performance in previous research are necessary but not sufficient conditions for reaching a certain level of academic achievement. Hoping to also provide new and relevant findings for the current context that add to previous research [15], we performed this test in a sample of students enrolled in a face-to-face university (*n* = 159) and a sample of students enrolled in an online university (*n* = 76). 

Specifically, the example sought to test: (a)The relationship between previous academic performance (as measured by the admission test), time spent studying, and conscientiousness as a personality trait with current academic performance among students enrolled in a face-to-face and an online university based on traditional methods (i.e., Pearson’s *r*).(b)If students with poor grades on the admission test, less time devoted to studying, and lower conscientiousness scores were precluded from high academic achievement in one environment (i.e., face-to-face or online) through NCA. We also wished to investigate whether a high ranking on the independent variables made the outcome possible in one or the other university setting, quantifying these effects.

We tested the sufficiency logic between the proposed variables through traditional methods and the necessity logic using NCA. The ultimate aim is to interpret the results obtained from both analyses and propose this combination as a way to overcome current limitations found for the use of only one technique [1,2] or the other [23,24].

## 2. Materials and Methods

### 2.1. Participants

Data were collected from students enrolled in Psychology degrees at face-to-face and online universities. A total of 245 students completed the survey (163 from the face-to-face university and 82 from the online university). One respondent was removed because of inconsistent data (i.e., indicating a GPA that would not allow access to the course being taken). Based on recent recommendations [15], we checked all variables for skewness, obtaining a value of 2.06 for the variable “time spent studying”. Since values over 2.0 are considered a sign of a highly skewed distribution [26], outliers (≥3 *SD* away from the mean) were removed. These corresponded to participants who reported over 40 h of study per week (*n* = 9), who were excluded aiming to prevent overestimating the NCA effect size [15]. 

The final sample comprised 235 college students (72.97% female, 67.66% enrolled in the face-to-face university), ranging from 18 to 67 years old (*M* = 25.05, *SD* = 8.15). Students enrolled in the online university were older (*M* = 33.62, *SD* = 9.45) than their counterparts in the face-to-face university (*M* = 20.96, *SD* = 4.52; *t*(235.32) = 42.33, *p* < 0.001). The number of students who lived independently (i.e., on their own, with their partner or own families, or with peers) compared to those living either with their family of origin or a foster family also differed significantly (χ^2^(1) = 56.72, *p* < 0.001). More students enrolled in the online university were also working as compared with students attending the face-to-face university (χ^2^(1) = 15.815 *p* < 0.001). 

### 2.2. Instruments

Since NCA could find notable effects when the distribution of the independent and the dependent variable produce empty space that is not relevant to a necessary but not sufficient relation [15,25], we only included the study predictors that have been consistently reported as correlated with our outcome and which are theoretically meaningful for the aims of our example. We tested the potentially necessary but not sufficient condition of the previous achievement [27,28,29] (as measured by the admission grade), time spent studying [30,31], and conscientiousness personality trait [32,33,34] on current academic performance. All of these predictors have also been found to be linked to academic achievement in online universities [35,36,37]. 

Participants reported their average admission grade, their GPA (both scored on a scale from 0 to 10) and how many hours per week they spent studying regularly. Conscientiousness (defined as a tendency to be self-controlled, perseverant, and highly motivated in goal-directed behaviours) was measured with the corresponding subscale of the Big Five Inventory (BFI, [38]), which comprises nine items (e.g., “I am a trustworthy, compliant worker”), rated on a 5-point agreement Likert-type scale that ranged from strongly disagree to strongly agree. Cronbach’s alpha was 0.78.

### 2.3. Procedure

The study was performed under the ethical standards of the Helsinki Declaration and the recommendations of the Ethics Committee of the University of Barcelona. For a self-report research with adult college students, the approval by Ethics Committee was not required at the time the study was conducted. Participants were psychology undergraduates recruited using convenience sampling. Students enrolled in the Psychometrics course were asked to respond to the questionnaires -hosted online by Qualtrics (www.qualtrics.com, accessed on 13 April 2020) platform and invite people they knew to participate. Participants were informed about the nature and objectives of the research before giving written consent; participation was voluntary, and data would remain confidential. Sample recruitment extended from September 2014 through April 2015.

### 2.4. Data Analysis

Total missingness in the whole dataset was 1.05%. As in previous studies, missing values were imputed with the mean of the variable for which missing data points were found [15]. Descriptive statistics were calculated for the outcome (i.e., GPA) and each independent variable (i.e., admission grade, hours per week spent studying, and conscientiousness score) for students enrolled in the face-to-face (*n* = 159) and online (*n* = 76) university. We then computed correlations among these variables and compared the groups through the *t*-test.

Two separate NCAs (i.e., face-to-face and online university) were performed. The outcome variable was GPA, while admission grade, hours of study per week, and conscientiousness were treated as independent variables to assess their status as necessary but not sufficient conditions. Two ceiling lines for each group of students were obtained. Following recommendations [22], the ceiling envelopment–free disposal hull (CE-FDH) has been conceived as appropriate for categorical data, as it produces a step-function ceiling line. The ceiling regression–free disposal hull (CR-FDH) is appropriate for continuous data and creates a regression line by joining the dots of the CE-FDH line. The percentage of observations that turn out to be below the line is used to describe the ceiling line accuracy. The CE-FDH line leaves all observations on or below the line by definition (100% accuracy), whereas CR-FDH lines aim to include the majority of observations below the line. If the CR-FDH ceiling line is below 95%, it should not be considered appropriate for interpretation, and the other ceiling line should be used to evaluate the necessary condition effect size. The significance of effect sizes was also tested, interpreting significant *p*-values (<0.05) as indicating that the effect size could not be randomly obtained using unrelated variables [39]. Outcome and condition inefficacy were also reported. Plots and bottleneck coefficients were also obtained for variables that showed a significant effect size. 

The database, complete code in R, and all the scatterplots with their respective ceiling lines and zones are available at https://osf.io/64jfq/ (accessed on 11 May 2022). In addition to the analyses made to each subsample (i.e., students enrolled in the face-to-face university vs students enrolled in the online university), we ran these analyses with the total sample. The results are included in the mentioned OSF profile as Appendix A. 

## 3. Results

### 3.1. Results Obtained through Classical Correlation Analyses

Table 1 shows the descriptive statistics, correlations, and t-test values for all variables of interest for students enrolled in face-to-face and online universities. Conclusions that can be extracted by relying on the traditional analyses are that students attending the face-to-face university had significantly higher levels of admission grades compared with students enrolled in the online university. By contrast, students from the online university reported significantly higher GPAs, time spent studying, and conscientiousness than their face-to-face university counterparts. Admission grades and conscientiousness were positively correlated with GPA, with slightly higher values for students enrolled in the online university compared with those attending face-to-face university. The two groups differed significantly in the outcome and all the predictors, which provides an interesting scenario with possible explanations of how results might diverge in each setting.

### 3.2. Example of Results Obtained Using NCA

Results of the NCA analyses are shown in Table 2. It can be seen that the level of accuracy of the CR-FDH ceiling line meets Dul’s criterion for interpretation (>95%) for students enrolled in the face-to-face university, but none of the predictors showed a significant effect. For students enrolled in the online university, only admission grades met the criterion that makes the CR-FDH ceiling line appropriate for interpretation; hence, the other predictors were interpreted using the CE-FDH ceiling line. Hours spent studying (*d* = 0.09, *p* = 0.01) and conscientiousness (*d* = 0.25, *p* < 0.001) yielded statistically significant effect sizes.

We can also see that inefficiency for study time among students enrolled in the online university condition is 50.00%. This means that study time beyond the median (i.e., 18 h per week) is not necessary for the highest possible level of GPA. However, conscientiousness inefficiency is 4.76%, which means a score above the 95.24% level of this predictor is no longer necessary to reach the highest possible degree of GPA. Outcome inefficiency is 0%; there is no outcome level in which these predictors are not necessary. 

Plots and bottlenecks for the predictors that had a significant effect are shown in Table 3. To achieve a GPA above the 70%, a student enrolled in the online university needed to spend a minimum of 23.6 h per week studying and score at least 46.8 on conscientiousness.

## 4. Discussion

Considering the current debates regarding the use and interpretations of statistical analyses frequently used in social sciences [1], [3], the present work aimed to provide an illustrative example of additional tools to complement the regular use of Pearson’s *r* coefficient within the framework of the social sciences. We expected to achieve our aim by illustrating NCA applied to two groups of university students enrolled in face-to-face and online universities. These groups provide real-case scenarios with plausible situations that researchers may find when applying NCA. We hope to help future users of these techniques to apply them after being inspired by the present example.

### 4.1. Descriptive and Correlation Analyses

If we were to use Cohen’s criteria [4], we could conclude that the effect of the previous achievement (as measured by admission grade) on current academic performance is small for students enrolled in the face-to-face university and medium for students enrolled in online universities. As recent studies have pointed out, these benchmarks are hard to understand or translate into actual effects [1,2]. However, we could put these findings into perspective if we consider that the average links between independent and dependent variables within the social psychology field are around *r* = 0.20 [6]. Hence, we could conclude that admission grade is important to predict current academic performance in both settings, although in students enrolled in online universities, this seems to be as strong as the tendency of men to weigh more than women [1]. In contrast, it seems to be a little below the average for their counterparts from the face-to-face university. 

When looking at the same coefficients to measure the effect of conscientiousness personality trait on academic performance, we could use an even more specific framework, following both traditional and recent recommendations [1,5]. We could use as benchmarks the percentiles reported in a recent study [2] to conclude that the link between this personality trait and our outcome among students enrolled in the face-to-face university is at the 70th percentile of all correlational effects reported for personality, whereas it is located right above the 80th percentile for students enrolled in an online university. As it can be seen, the link looks much stronger (and, in our opinion, real) than when relying on classical guidelines for both settings, although the effect is still greater in online universities.

However, these analyses do not tell us if any of the analysed predictors (admission grade, study time, or conscientiousness) are needed to achieve a certain level of GPA and to what degree. Besides, they are still hardly applicable or valuable to design interventions to improve academic performance. 

### 4.2. NCA Results

The NCA can complement these results by adding details about how much of a variable X is needed to allow Y to occur. As previously mentioned, the NCA can be applied to determine if a certain minimum GPA needs a specific level of admission grade, study hours per week, or conscientiousness. It also can be applied to many other research questions in social sciences that are aimed at detailing the link between very well-known associated variables, such as achievement and earnings [40,41] or social support and mental health [42,43]. Through NCA, we could look for answers to questions such as: is there a minimum level of education needed to be capable of earning a certain amount of money per year? How much social support is needed to score higher than the mean in subjective well-being? Hence, the following conclusions are to be interpreted as part of an illustrative example that may guide future uses of NCA in the social sciences field. 

In our example, the NCA revealed a non-significant effect of admission grade and conscientiousness on the students’ GPA attending the face-to-face university. This might suggest that a minimum admission grade and a certain degree of conscientiousness are indeed necessary to achieve a certain level of GPA in this setting, although the effect was not distinguishable from chance in our sample. By contrast, the effect of time spent studying and conscientiousness was significant among students enrolled in the online university. These results suggest that conscientiousness may be a necessary-but-not-sufficient condition for achievement among students enrolled in online universities, i.e., a certain level of conscientiousness is needed for it to be possible to achieve a certain level of GPA. Stating that a predictor X is needed for outcome Y to occur represents additional information to quantify the link reported by the correlation coefficient. 

From the findings reported in this example, we could also conclude that study time is necessary to achieve a certain degree of GPA. Thanks to the condition inefficiency cut-off point, we could also conclude that more than 18 h per week studying becomes pointless in the pursuit of increasing your level of GPA. Students may easily use this information to improve their performance: if they spend less than 18 h studying, they can realize that more time is needed to achieve a higher GPA; if they spend this time studying or more, they can start assessing whether their studying techniques are effective or if they are managing their time efficiently [44]. Our example is also unique since it shows that at least a minimum value greater than 0 is needed for both predictors, i.e., no level of GPA is possible to achieve without studying at a conscientiousness score higher than nine. Bottlenecks provide additional information to target, even more specifically, the students’ aims, as they are useful for defining the extent to which predictors are needed to reach the desired level of the outcome. Making publicly available the bottlenecks identified in the present example regarding study time and conscientiousness may be useful for students to consider how to raise their GPA based on their current levels on these variables and their personal objectives. For example, students enrolled in online universities with the aim of obtaining a GPA over the 90th percentile can check if they are spending enough time studying since, according to the bottlenecks, we know at least a minimum of 31.3 h per week is needed to achieve a GPA this high. If they are already spending this time or more, they can check their conscientiousness score, as, according to our results, at least a score of 59.6 is needed to achieve this level of GPA. If they score less, they could try developing more learning strategies related to being a conscientious person, such as postponing instant self-gratification to pursue a longer-term goal. Furthermore, if they have none of these problems, they can focus on something else, such as how distracted they get during online homework [45] or strategies to reinforce effective study habits [44]. Thus, both inefficiencies and bottlenecks provide intuitive information that is understandable and easily translated to concrete actions, as suggested in recent studies [10].

### 4.3. Limitations

In summary, NCA could complement with valuable and applicable information the conclusions we usually arrive at when relying on Pearson’s coefficients or other types of effect sizes. In addition to previous proposals [10,11], NCA can represent a new tool to translate research results to their actual consequences in concrete, real-life situations. However, NCA does not respond to all of the aspects pointed out as “challenging” in previous studies, arguing the need for new ways of making conclusions from our results. The use of these techniques is still unfamiliar to most researchers; therefore, as pointed out in previous works [1], it will probably take a long time until researchers start to use these new indicators more intuitively. However, as more research includes NCA in their method of analysing, more meaningful comparisons will be possible. 

Another important flaw is that the criteria to qualify an effect size as small, medium, or large within the NCA framework [13] is also based on standardized cut-off points, which probably neglects contextual aspects that affect the interpretation. That is why the results of our examples were not classified using this benchmark. However, we could say that our findings are consistent with previous research using NCA in samples of students enrolled exclusively in face-to-face universities [15]. They are also in line with a recent study proposing conscientiousness as a necessary but not sufficient condition for avoiding certain impulsive behaviours [16]. As previously suggested, we expect that over time, as the social sciences use this type of methods more widely, the reference point for NCA estimates could be built, and all members of the scientific community will become more familiar with them. 

It is also important to borne in mind some practical aspects when using NCA. First of all, the same assumptions regarding reliability as when using exclusively other effect sizes coefficients have to be taken into account [1,39]. Specifically, although some authors have argued that 25–30 observations were sufficient to compute meaningful correlations [46], others have proposed a minimum sample size of 250 [47] or an estimation of the sample size depending on several study factors [48,49]. Even though we carefully checked that the assumptions of the statistical techniques used were met and the demographic differences in our subsamples made it worth analysing them separately, the link between effect sizes (both NCA and Pearson’s *r*) should be considered when interpreting the results. We should also recall warnings made by recent simulation studies proposing some limitations to identify necessity in large samples [23]. How to handle the effect of atypical values on the empirical scope [15] should also be further tested. For the moment, the only way to ensure that NCA is applicable and suitable for our research is to consider the scope’s probability to reflect real-case scenarios or to stick to what has been defined as theoretically possible. In our example, study hours per week ranged from 2 to 100, so we had to proceed to remove participants that reported a nearly impossible number of hours spent studying per week. 

Finally, even though it is beyond the scope of this article, it is worth considering controversy about the way in which significance can be manipulated in NCA analyses [23] and proposals that seem to outperform NCA in some cases, such as quality condition analyses [24]. Although Dul and his team have replied to these arguments [50,51], future studies providing examples of how to use these techniques and how their results might be useful to overcome the limitations of the currently used tools will be relevant to pursue the debate in our research field. 

There are also some limitations worth mentioning about the data we used for our example. First of all, this was cross-sectional and obtained purely through self-report, so effects such as the attributional process [15] may have influenced students’ responses. Although we tried to address this issue with a proper instrument design, this limitation should be borne in mind when interpreting the results. A second and related issue concerns sample representativeness and the generalizability of results since the sampling procedure most likely favoured the recruitment of more motivated students. The sample size might also influence the estimates obtained for traditional and NCA analyses, so future research with larger samples should replicate examples such as ours. Finally, we have compared online and face-to-face settings based on real-world experiences and not through an experimental design. Even though our study provides findings with high ecological validity, the approach used should be considered when interpreting the results. 

## 5. Conclusions

Traditional effect sizes are rarely reported or misleadingly interpreted. The NCA could complement these widely used coefficients, as it can be helpful to determine if a variable is needed for an outcome to occur and to what extent. In our illustrative example, a certain level of conscientious personality trait and a minimum number of hours per week spent studying seem to be necessary to achieve a minimum expected level of GPA for students enrolled in online universities. This information complements the traditional association between these variables (i.e., study time and conscientiousness) and the dependent variable (i.e., achievement measured through GPA), providing the opportunity to design targeted interventions. We would like to encourage researchers through this study to use new benchmarks to interpret their classical analyses and apply NCA as an additional informative technique when they consider it appropriate.

## Figures and Tables

**Figure 1 ijerph-19-09402-f001:**
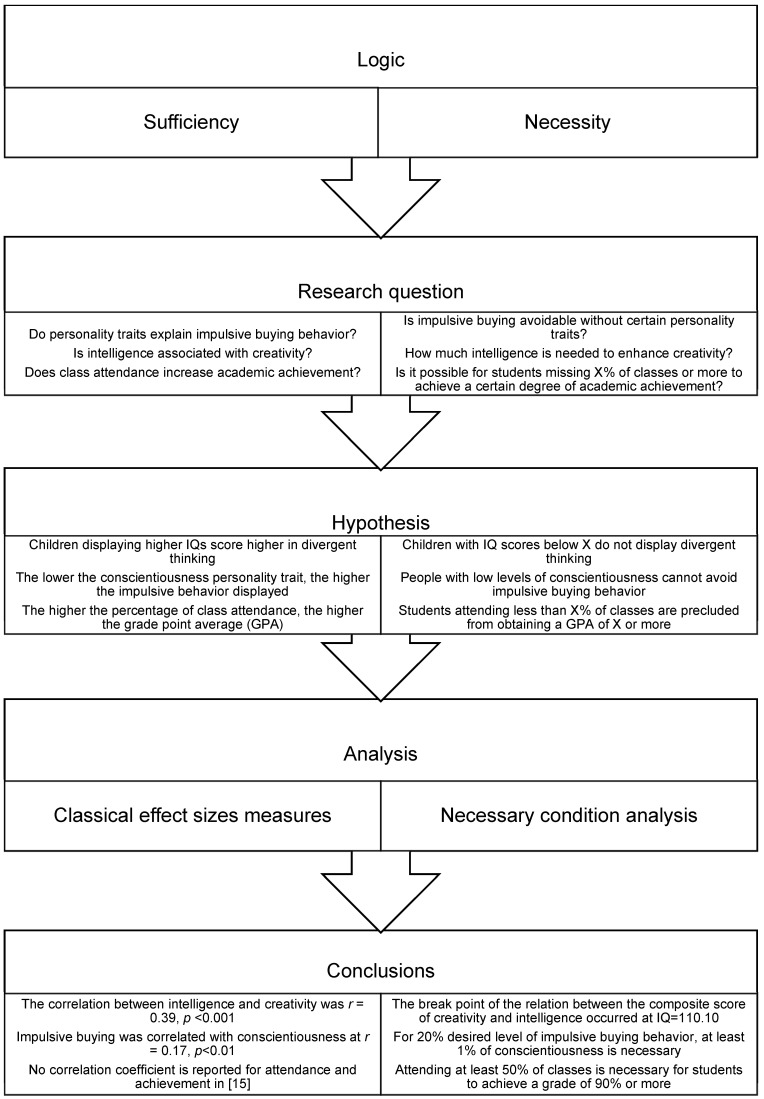
Examples of sufficiency and necessity logic are extracted from [15,16,17], respectively.

**Table 1 ijerph-19-09402-t001:** Descriptive statistics, bivariate correlations, and t-test values.

	Students Enrolled in the Face-to-Face University						Students Enrolled in the Online University						
	Mean	SD	Range	1.	2.	3.	Mean	SD	Range	1.	2.	3.	*t*
1.	6.80	0.72	5.5–9.20				7.29	0.79	5.0–8.76				−4.20 ***
2.	7.80	0.71	6.00–9.36	0.18 *			6.90	0.96	5.0–9.60	0.39 ***			7.25 ***
3.	12.65	8.62	1–40	0.13	0.02		18.59	10.76	2–100	0.08	0.06		−4.21 ***
4.	31.87	5.54	17–44	0.26 ***	0.04	0.15	35.17	4.93	24–45	0.32 **	0.19	0.13	−4.61 ***

*Note.* 1. Grade point average (GPA), 2. Admission grade, 3. Study time, 4. Conscientiousness. * *p* < 0.05, ** *p* < 0.01, *** *p* < 0.001.

**Table 2 ijerph-19-09402-t002:** Necessary condition effect sizes and significance tests for admission grade, study time, and conscientiousness as predictors of college grade point average (GPA).

	CE-FDH									CR-FDH	
	ES	*p* Value	A	OI	CI	ES	*p* Value	A	OI	CI	Skewness
Students enrolled in the face-to-face university											
1.	0.29	0.002	100%	27.03%	31.55%	0.20	0.095	98.10%	38.84%	36.40%	−0.07
2.	0.07	0.209	100%	40.54%	51.28%	0.08	0.115	98.70%	62.08%	57.30%	0.90
3.	0.17	0.427	100%	10.81%	44.44%	0.16	0.357	98.10%	34.54%	50.84%	−0.27
Students enrolled in the online university											
1.	0.08	0.203	100%	26.60%	43.48%	0.10	0.051	92.10%	57.54%	51.51%	0.40
2.	0.09	0.010	100%	0.00%	50.00%	0.11	0.048	80.30%	42.08%	62.18%	0.34
3.	0.17	0.004	100%	0.00%	4.76%	0.21	0.000	82.90%	42.02%	28.02%	−0.07

*Note.* 1. Admission grade, 2. Study time, 3. Conscientiousness. ES = Effect size, A = Accuracy, OI = Outcome inefficiency CI = Condition inefficiency CE-FDH = ceiling envelopment–free disposal hull; CR-FDH = ceiling regression–free disposal hull. *p* values were estimated with 10,000 permutations and are treated as significant if *p* < 0.05, considering the threshold proposed by Dul et al. (2020). Accuracy refers to the percentage of values that are below the CR-FDH ceiling line. GPA skewness was 0.73 for face-to-face university and −0.40.

**Table 3 ijerph-19-09402-t003:** Bottlenecks for study time and conscientiousness for students enrolled in the online university.

	Students Enrolled in the Online University (*n* = 81)	
	Ceiling Line y = 0.15x + 6.27	Ceiling Line y = 0.14x + 3.12
GPA	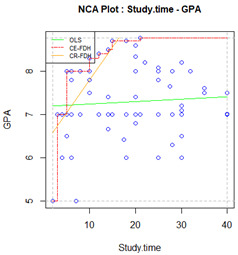	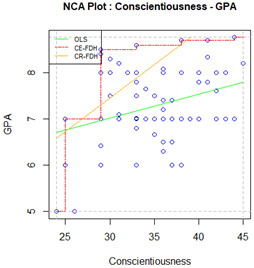
GPA (%)	Study time	Conscientiousness
0	NN	NN
10	NN	NN
20	NN	NN
30	NN	NN
40	NN	NN
50	5.2	9.9
60	11.7	22.3
70	18.2	34.7
80	24.8	47.1
90	31.3	59.6
100	37.8	72.0

*Note.* GPA: grade point average. NN = “not necessary”.

## Data Availability

All data and code to reproduce findings reported in this study may be found at https://osf.io/64jfq/ (accessed on 11 May 2022).

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
