# Peer review of "Proposing Necessary but Not Sufficient Conditions Analysis as a Complement of Traditional Effect Size Measures with an Illustrative Example"

_ijerph, 2022, doi:10.3390/ijerph19159402_

Round 1

Reviewer 1 Report

The manuscript “Proposing Necessary-But-Not-Sufficient Conditions Analysis as a Complement of Traditional Effect Size Measures with an Illustrative Example” addresses an important topic related to psychometrics and applicable to psychological, environmental and public health research more widely.

The manuscript is generally well written and easy to follow.

1.       Figure 1 is very helpful in explaining the relevant concepts. However, it contains several typos and grammatical errors that should be remedied

2.       On page 5, line 181, it’s unclear what “As it can be drawn” means. Please elaborate or rephrase.

3.       The statement on ethics (“The study was performed under the ethical standards of the Helsinki Declaration and the recommendations of the Ethics Committee of the BLINDED FOR REVIEW.”, p. 6) is unclear. Was ethics approval obtained for the study? Or was the study exempt? Please provide a clear description of ethical clearance.

4.       The samples from in-person and online universities differed in terms of demographics and outcomes, which makes separate analyses worthwhile. One concern I have is the relatively small sample sizes resulting from that split; for example, a sample size of at least 250 is usually recommended for correlations (https://doi.org/10.1016/j.jrp.2013.05.009). It would hence be helpful to also conduct the comparisons across the full sample to see whether the higher stability of the obtained coefficients changes or confirms the results.

5.       In the Table 2 notes, “Conscienciousness” should read “Conscientiousness”.

6.       Only one of the words (“whether” or “if”) are needed in the sentence on page 10 (“…can check whether if they are…”)

Author Response

The manuscript “Proposing Necessary-But-Not-Sufficient Conditions Analysis as a Complement of Traditional Effect Size Measures with an Illustrative Example” addresses an important topic related to psychometrics and applicable to psychological, environmental and public health research more widely.

The manuscript is generally well written and easy to follow.

Response: Thank you very much for your positive feedback.

1. Figure 1 is very helpful in explaining the relevant concepts. However, it contains several typos and grammatical errors that should be remedied

Response: Thank you very much for pointing this out. We have now corrected the texts in the figure. Since the “Track changes” function does not track corrections inside the figure, we transcribed the corrections made in a table (see attached).

2. On page 5, line 181, it’s unclear what “As it can be drawn” means. Please elaborate or rephrase.

Response. Thank you very much for your observation. We agree it was not an appropriate connector, so we removed it from the sentence.

3. The statement on ethics (“The study was performed under the ethical standards of the Helsinki Declaration and the recommendations of the Ethics Committee of the BLINDED FOR REVIEW.”, p. 6) is unclear. Was ethics approval obtained for the study? Or was the study exempt? Please provide a clear description of ethical clearance.

Response: Thank you again. We added the phrase “For a self-report research with adult students, the approval by Ethics Committee was not required at the time the study was conducted” next to the mentioned text.

4. The samples from in-person and online universities differed in terms of demographics and outcomes, which makes separate analyses worthwhile. One concern I have is the relatively small sample sizes resulting from that split; for example, a sample size of at least 250 is usually recommended for correlations (https://doi.org/10.1016/j.jrp.2013.05.009). It would hence be helpful to also conduct the comparisons across the full sample to see whether the higher stability of the obtained coefficients changes or confirms the results.

Response: Thank you for this suggestion. As proposed, we ran the analyses with the complete dataset. Correlations between two of the independent variables (i.e., admission grade and study time) and the outcome (i.e., GPA) differed. In terms of NCA effect sizes, only the results of admission grade as a necessary condition in the face-to-face differed with respect to the full dataset. We provided all this information as supplementary material, should they may complement the interpretation of the results (see page 7). We also included the following statement mentioning this in the limitation section:

“Specifically, although some authors have argued that 25-30 observations were sufficient to compute meaningful correlations [47], others have proposed a minimum sample size of 250 [48] or an estimation of the sample size depending on several study factors [49], [50]. Even though we carefully checked that the assumptions of the statistical techniques used were met and the demographic differences in our subsamples made it worth analyzing them separately, the link between effect sizes (both NCA and Pearson’s r) should be considered when interpreting the results.”

5. In the Table 2 notes, “Conscienciousness” should read “Conscientiousness”.

Response: Thank you very much, we corrected the spelling mistake.

6. Only one of the words (“whether” or “if”) are needed in the sentence on page 10 (“…can check whether if they are…”)

Response: Thank you once more. We corrected the error in both sentences in page 10.

Thank you very much for your keen eye and helping us improving the manuscript.

Reviewer 2 Report

The authors provide background and an illustrated example of necessary condition analysis (NCA) along with a comparison to the more commonly used correlation analyses. They posit that NCA can serve as an effect size measure that would be more meaningful than the somewhat arbitrary criteria like Cohen's d or Pearson's r.

They provide an overview of NCA and then illustrate its use, as well as a comparison with correlation analysis, in an example examining academic performance as a function of previous achievement, time spent studying, and conscientious personality trait.

Overall, I found the paper well organized and presenting a clear description of the technique and its relation to traditional statistical approaches. As NCA originated in the domain of organizational research it is not as well known in other areas, particularly in the social sciences. This paper will serve useful in disseminating the technique more widely.

I have no specific issues with the manuscript as-is, with the exception of reference 6 - I believe the year should be 1998 rather than 1888.

Author Response

Dear reviewer,

Thank you very much for all your appreciations. We also expect this work will serve to furtherly disseminate the use of this technique. We have now corrected the year of the reference, thanks for pointing it out.

Best regards,

The authors.
